# Cell-Free Protein Synthesis Using S30 Extracts from *Escherichia coli* RFzero Strains for Efficient Incorporation of Non-Natural Amino Acids into Proteins

**DOI:** 10.3390/ijms20030492

**Published:** 2019-01-24

**Authors:** Jiro Adachi, Kazushige Katsura, Eiko Seki, Chie Takemoto, Mikako Shirouzu, Takaho Terada, Takahito Mukai, Kensaku Sakamoto, Shigeyuki Yokoyama

**Affiliations:** 1RIKEN Systems and Structural Biology Center, Yokohama 230-0045, Japan; jiro.adachi@riken.jp (J.A.); kazushige.katsura@riken.jp (K.K.); eiko.seki@riken.jp (E.S.); chie.takemoto@riken.jp (C.T.); mikako.shirouzu@riken.jp (M.S.); tera@riken.jp (T.T.); takahito.mukai@rikkyo.ac.jp (T.M.); kensaku.sakamoto@riken.jp (K.S.); 2Division of Structural and Synthetic Biology, RIKEN Center for Life Science Technologies, Yokohama 230-0045, Japan; 3RIKEN Structural Biology Laboratory, Yokohama 230-0045, Japan; 4Laboratory for Cell-Free Protein Synthesis, RIKEN Center for Biosystems Dynamics Research, Osaka 565-0874, Japan; 5Laboratory for Protein Functional and Structural Biology, RIKEN Center for Biosystems Dynamics Research, Yokohama 230-0045, Japan; 6RIKEN Cluster for Science, Technology and Innovation Hub, Yokohama 230-0045, Japan; 7Department of Life Science, College of Science, Rikkyo University, Tokyo 171-8501, Japan; 8Laboratory for Nonnatural Amino Acid Technology, RIKEN Center for Biosystems Dynamics Research, Yokohama 230-0045, Japan

**Keywords:** cell-free protein synthesis, non-natural amino acid, release factor 1, RF-1, 3-iodo-l-tyrosine, RFzero-iy

## Abstract

Cell-free protein synthesis is useful for synthesizing difficult targets. The site-specific incorporation of non-natural amino acids into proteins is a powerful protein engineering method. In this study, we optimized the protocol for cell extract preparation from the *Escherichia coli* strain RFzero-iy, which is engineered to lack release factor 1 (RF-1). The BL21(DE3)-based RFzero-iy strain exhibited quite high cell-free protein productivity, and thus we established the protocols for its cell culture and extract preparation. In the presence of 3-iodo-l-tyrosine (IY), cell-free protein synthesis using the RFzero-iy-based S30 extract translated the UAG codon to IY at various sites with a high translation efficiency of >90%. In the absence of IY, the RFzero-iy-based cell-free system did not translate UAG to any amino acid, leaving UAG unassigned. Actually, UAG was readily reassigned to various non-natural amino acids, by supplementing them with their specific aminoacyl-tRNA synthetase variants (and their specific tRNAs) into the system. The high incorporation rate of our RFzero-iy-based cell-free system enables the incorporation of a variety of non-natural amino acids into multiple sites of proteins. The present strategy to create the RFzero strain is rapid, and thus promising for RF-1 deletions of various *E. coli* strains genomically engineered for specific requirements.

## 1. Introduction

Cell-free protein synthesis has become the preferred method for protein preparations, as it offers numerous advantages over bacterial and eukaryotic cell-based expression method [1]. The open nature of an in vitro system facilitates modifications and optimizations of reactions, and it enables the synthesis of difficult proteins, such as physiologically toxic proteins, integral membrane proteins, and large protein complexes, by allowing supplementation with multiple additives, such as chemicals, detergents, lipids, and molecular chaperones. For example, large amounts of membrane proteins are successfully generated by cell-free systems with detergents and lipids as additives [2]. Precise stable isotope labeling of proteins, which are inevitably metabolized upon in vivo expression, could be achieved by using specific inhibitors of intracellular metabolic reactions [3].

The *Escherichia coli* cell extract-based cell-free protein synthesis is one of the most practical and efficient cell-free systems, and thousands of proteins have been synthesized for functional and structural studies [1], and for pharmaceutical development. Methods for engineering the *E. coli* genomic DNA are now well established, and thus the development of novel cell extracts for cell-free protein synthesis has become possible [4,5].

The site-specific incorporation of non-natural amino acids into proteins has become an important technology for protein engineering. More than 100 non-natural amino acids have been site-specifically incorporated into proteins for various purposes, such as conjugations with fluorescent probes, polymers, and drugs [6]. Some amino acids generated by protein post-translational modifications can be translationally incorporated into proteins to make them directly in the modified states [7]. Aminoacyl-tRNA synthetase (aaRS) and tRNA pairs from bacteria and archaea, which are orthogonal to or not recognized by the endogenous aaRS and tRNA pairs are used. However, non-natural amino acids are sometimes toxic or poor in cellular uptake and are fundamentally difficult to incorporate into proteins by in vivo protein expression methods. The excess amounts of the orthogonal tRNA and aaRS pair for non-natural amino acid incorporation also exhibit cellular toxicity during in vivo expression. Therefore, cell-free protein synthesis is a suitable technology for site-specific incorporations of non-natural amino acids, not only for its ability to synthesize difficult proteins as mentioned above, but also for its non-cellular nature [8,9].

The amber (UAG) stop codon is commonly reassigned as a target codon to incorporate a non-natural amino acid during translation, while it is normally recognized as a translation termination signal by release factor 1 (RF-1) in *E. coli*. To improve the incorporation efficiency of a cell-free system using the UAG codon, various strategies to minimize the RF-1 activity in the cell extract have been attempted, such as by adding RF-1-inactivating RNA aptamer to the reaction mixture [10], reconstituting the cell-free system without RF-1 [11], and engineering the RF-1 for tag-mediated removal from the extract [12]. However, despite these manipulations, low levels of RF-1 activities often remain within these cell extracts.

Several RF-1-free *E. coli* strains have been developed by deleting the RF-1-encoding gene *prfA* from the genomic DNA [13,14,15,16], giving rise to another strategy, to use the cell extracts of such RF-1-free *E. coli* strains for protein synthesis. The *E. coli* genome originally harbors over 300 genes ending with UAG stop codons, and thus the simple deletion of *prfA* or the disabling of the RF-1 function severely affected the growth of *E. coli*, resulting in low protein productivity. In contrast, the replacement of a large number of UAG codons with other stop codons allowed the deletion of the essential *prfA* gene, with minimal effects on the growth and protein production. Using the cell extracts from these strains, several cell-free protein synthesis methods exhibited high incorporation efficiencies [17,18,19]. B-60.ΔA::Z and B-95.ΔA are RF-1-free *E. coli* strains, with genomes in which 60 and 95 UAG codons were mutated, respectively [16]. Using cell extracts from these strains, we have reported cell-free protein synthesis with multiple site-specific incorporations of non-natural amino acids [20].

Meanwhile, many *E. coli* strains have been generated specifically for various purposes, by deleting multiple enzymes or metabolic pathways. Some of these strains have been designed to incorporate non-natural amino acids. As an example, for *O*-phosphotyrosine incorporation, five genes for endogenous phosphatases were deleted from the *E. coli* genome [21]. To utilize the functions of these diverse *E. coli* strains, the methods to create their RF-free versions, by replacing a large number of UAG codons, require extensive gene editing and laborious work. Similarly, the additions of specific properties to these UAG codon-replaced RF-free *E. coli* strains also require multiple genome editing steps.

Therefore, we have generated a rapid method to create a new RF-1-free *E. coli* strain, RFzero, with only a minimal decrease of the *E. coli* growth rate [22]. This strategy, involving the transformation of a BAC plasmid harboring seven coding sequences for deleting *prfA*, is quite rapid and simple, as compared to the above strategies with numerous genomic engineering steps. Only one mutation of a gene is required to create RF-1-free *E. coli* strains from diverse *E. coli* strains. In practice, we have generated RFzero strains from different *E. coli* strains, including the *E. coli* K-12 strains BW25113, W3110, and HMS174(DE3) and the *E. coli* B strain BL21(DE3), for various purposes [23,24]. The RFzero-iy strain was generated for incorporating 3-iodo-l-tyrosine (IY) [25], with the expression of a previously developed *Methanococcus jannaschii* tRNA^Tyr^ variant and a tyrosyl-tRNA synthetase variant, iodoTyrRS. When this RFzero-iy strain was cultured with IY, over 300 gene transcripts of the *E. coli* genome that end with UAG stop codons were translated to IY, and thus, the protein products have several additional residues at the C-termini. These translations prevent ribosomes from stalling at the UAG sites, which should have arisen from the deletion of RF-1. Thus, the RFzero-iy strain recovered vigorous growth when it was cultured with IY [25]. Using the RFzero-iy strain, other non-natural amino acids could be incorporated into proteins, by supplementing the media with specific non-natural amino acids during the culture, although these non-natural amino acids showed decreased protein productivities [23]. Our preliminary trial using the cell extract from the RFzero BL21(DE3) strain incorporated four acetyllysines into histone H4, with high efficiency [23].

In this study, we present the precise strategy for the site-specific incorporation of various non-natural amino acids using this RFzero extract-based cell-free system. The cell extracts were created from the RFzero-iy strains. We describe the detailed method for the cell extract preparation to overcome the distinctive difficulties of using the RFzero strain, and demonstrate the creation of two RFzero-iy strains from different *E. coli* strains. We confirmed the incorporations of non-natural amino acids (Figure 1A) with high efficiency and excellent productivity by cell-free protein synthesis, using a single type of cell extract (Figure 1B). The incorporation was not restricted to the use of TyrRS variants, but it was applicable to tRNA^Pyl^/PylRS variants pairs, simply by their supplementation into the reaction solution. Together with the rapidity of constructing new RF-1-free strains from *E. coli* strains specifically generated for various purposes, our cell-free protein synthesis strategy will expand the adaptation of these diverse *E. coli* strains to incorporate non-natural amino acids into proteins.

## 2. Results

### 2.1. Comparison of RFzero Strains as S30 Extract Sources

Since RFzero-iy exhibited the most vigorous growth among the variants of RFzero strains [23], we generated two types of RFzero-iy strains from an *E. coli* K-12 strain, BW25113, and an *E. coli* B strain, BL21(DE3). The growth rates of the generated strains were compared at 37 °C in LB medium and 2YT medium, which is richer in nutrients and commonly used for the preparation of S30 extracts. Both strains exhibited almost the same growth profiles in LB medium (Figure 2A), and the BW25113-based RFzero-iy reached a slightly higher cell density in 2YT medium (Figure 2B).

To confirm their utility as S30 extract strains, both strains were grown in 2YT at 37 °C for three hours and harvested before entering the late-log phase. As shown in Figure 2B, the cell densities at the harvest point were similar between the two strains, and comparable amounts of S30 extract were obtained from them. The small-scale dialysis-mode cell-free protein synthesis of the chloramphenicol acetyltransferase (CAT) protein was conducted with these S30 extracts, to confirm the protein productivity (Figure 2C). The BL21(DE3)-based RFzero-iy S30 extract produced a significantly larger amount of CAT, as compared to the BW25113-based RFzero-iy S30 extract. This result is consistent with the previous report showing that the S30 extract from a BL21 strain is more productive than the extract from a K-12 strain [26]. Thus, we decided to use the BL21(DE3)-based RFzero-iy strain in further experiments.

### 2.2. Optimization of S30 Extract Preparation from BL21(DE3)-based RFzero-iy

For the growth of RFzero-iy, supplementation of IY is required in the growth media [22]. However, if a significant level of IY from the media contaminates the S30 extract, then the extract would translate the UAG codon to IY, and the ability of the RFzero-iy extract would be unintentionally limited to IY. To minimize this risk, we attempted to reduce the concentration of IY, which is generally supplemented at 0.1 mg/mL. However, a reduction to one-tenth of this IY concentration hampered the bacterial growth and decreased the cell density by 80% at the late-log phase. Thus, we decided to continue supplementing IY at 0.1 mg/mL, and searched for other approaches.

Temperature and aeration rate are two other common factors affecting bacterial growth. Since the cultivation of *E. coli* at 30 °C reportedly enhances the productivity of the S30 extract [27], we decided to grow RFzero-iy at 30 °C in the subsequent experiments.

We then examined the effect of the aeration during the cell culture, using baffled flasks and a jar fermenter with different airflow rates (Figure 3A). The culture in the baffled flask reached a higher cell density than that the standard flask. For the jar fermenter, an airflow rate of 3.5 L/min considerably improved the cell growth, as compared to 0.5 L/min. Both results indicate that the growth of RFzero-iy is effectively enhanced with better aeration. Since the jar fermenter produced a comparable cell mass to that of the baffled flask, we decided to use the jar fermenter for the further preparation of large amounts of RFzero-iy S30 extracts.

### 2.3. Protein Productivity of S30 Extract from BL21(DE3)-Based RFzero-iy

To confirm the protein productivity of cell-free systems, using S30 extracts from RFzero-iy and its parent strain BL21(DE3), the S30 extracts were prepared as follows. Both strains were cultivated in the jar fermenters with the 3.5 L/min airflow rate at 30 °C. The growth rate of RFzero-iy was significantly slower than that of BL21(DE3) (Figure 3B). The RFzero-iy cells were harvested at 1.5 OD_600_ after 7 h, while the BL21(DE3) cells were harvested at 2.5 OD_600_ after a 4 h. The volume of the S30 extract obtained from a 1-l cell culture was 3.0 ml for RFzero-iy, corresponding to 57% of that of BL21(DE3).

The dialysis-mode cell-free protein synthesis of wild-type GFPS2 was conducted, using the produced S30 extracts in the presence and absence of IY (Figure 4). The protein productivity with the RFzero-iy-based S30 extract was about 60% of that of BL21(DE3). No significant differences were obtained with or without IY, using both S30 extracts. Thus, although the total productivity of the S30 extract was lower with RFzero-iy, the property of the S30 extracts corresponding to IY seems to be similar.

### 2.4. Incorporation of IY in Response to UAG in RFzero-iy-Based Cell-Free System

The dialysis-mode cell-free protein synthesis using the RFzero-iy-based S30 extract was conducted, to examine the incorporation efficiencies of IY at various UAG codon sites. This IY is a non-natural amino acid that has been incorporated into proteins, for purposes such as labeling and conferring heat resistance, and has actually been used for X-ray crystallographic analyses.

The tyrosine codons in pN11-GFPS2 at positions 21, 88, 192, 231, and 249 were individually replaced with UAG codons. With IY supplementation, the GFPS2 variants were efficiently synthesized, with productivities of more than 90% as compared to the amount of the wild-type GFPS2 (Figure 5A). These results indicate that the RFzero-iy-based S30 extract is highly effective for the site-specific incorporation of IY into various positions within the structure of the target protein.

Meanwhile without IY supplementation, hardly any full length GFPS2 variant proteins were synthesized and exhibited the fluorescence. These data also indicate that there was not significant contamination of IY in the RFzero-iy-based S30 extract from the growth media. Thus, the UAG codon of the template plasmid was kept unassigned in the RFzero-iy-based S30 extract, with neither RF-1 nor aminoacylated tRNA recognizing the UAG codon. Although the extract contains the iodoTyrRS and UAG-reading tRNA^Tyr^, the tRNA^Tyr^ was uncharged without IY in the cell-free reaction mixture, and was thus non-functional for protein synthesis. This result suggests that the RFzero-iy-based S30 extract has the potential to incorporate other non-natural amino acids, by supplementing various orthogonal tRNA and aaRS pairs into the cell-free protein synthesis solution (Figure 1B).

### 2.5. Reassignment of UAG to other Non-Natural Amino Acids in RFzero-iy-Based Cell-Free System

Many other archaeal TyrRS variants have been designed, such as TysRS for *O*-sulfo-l-tyrosine (Tys, also designated as SfY [23,28]), AzFRS for *p*-azido-l-phenylalanine (AzF), and *p*BpaRS for *p*-benzoyl-l-phenylalanine (*p*Bpa) [6]. Interestingly, PylRS exhibits unusually broad specificities for the substrates [29], and its designed variants accept multiple substrate amino acids. One of the variants, PylRS(R61K−Y384F−Y306A), successfully esterifies multiple lysine derivatives [30], such as *N*^ε^-benzyloxycarbonyl-l-lysine (ZLys) [31], and *N*^ε^- (*m*-azidobenzyloxycarbonyl)-l-lysine (*m*AzZLys) [32] (Figure 1A).

In the absence of IY, the supplemented TyrRS variants charge their specific tyrosine derivatives to the UAG-reading tRNA^Tyr^ within the RFzero-iy-based S30 extract, or the supplemented PylRS variant charges lysine derivatives to the supplemented tRNA^Pyl^, and the UAG codon is then reassigned for protein synthesis (Figure 1B). Using the combinations of AzFRS and AzF, TysRS, and Tys, *p*BpaRS, and *p*Bpa, PylRS(R61K−Y384F−Y306A) and ZLys, and PylRS(R61K−Y384F−Y306A) and mAzZLys, all of the aaRSs charged the non-natural amino acids, and the full length N11-GFPS2 variants were synthesized by the RFzero-iy-based cell-free system (Figure 5B). Among these non-natural amino acids, AzF and *p*Bpa were incorporated as efficiently as IY. The incorporation efficiency decreased in the order of ZLys, Tys, and mAzZLys. The incorporation site Y21 is located in the linker sequence of pN11-GFPS2, outside of the GFP, and therefore the effects of the incorporated non-natural amino acids on the fluorescent intensity are presumed to be small. On the other hand, the aminoacylation activities of aaRSs vary with each non-natural amino acid, and the incorporation rates of non-natural amino acids may vary during translation. Thus, these differences in the incorporation rates of the non-natural amino acids shown in Figure 5B are presumed to arise from the differences in the aaRS activities with each non-natural amino acid, and the differences in the incorporation rates of non-natural amino acids.

This result demonstrated that the UAG codon, in the unassigned codon state, could easily be reassigned to any other desired non-natural amino acid by supplementing the cell-free reaction mixture, using a single type of cell extract, with a specific aaRS variant and UAG-reading tRNA pair. Furthermore, the high incorporation efficiency at the UAG codon is expected to allow the incorporation of non-natural amino acids at multiple positions within target proteins.

## 3. Discussion

In this study, we developed a method to prepare cell extracts from *E. coli* RFzero strains for the site-specific incorporation of non-natural amino acids using the extract-based cell-free protein synthesis. The open nature of the cell-free system is fundamentally suitable for the protein incorporation of non-natural amino acids, which are sometimes toxic or poor in cellular uptakes. The deletion of the RF-1 activity from *E. coli* strains is the key for the efficient incorporation of the non-natural amino acids using UAG stop codons. Thus, this strategy, combining the cell-free synthesis method with an RF-1-free *E. coli* strain, is advantageous for a wide range of applications involving non-natural amino acid incorporation.

The incorporation of IY into several positions of GFPS2, by RFzero-iy-based cell-free protein synthesis, presented an interesting advantage. As a general rule of the codon context effect in *E. coli*, UAG codons followed by U or C are translated less efficiently than UAGs followed by A or G [33]. However, even though the tested UAG codon positions (21, 88, 192, 231, and 249) have neighboring codons with the first letters of U, G, A, C, and C, respectively, the effects of this codon context rule were not detected in the RFzero-based cell-free system (Figure 5A), and IYs were efficiently incorporated into all of the positions. Considering that the codon context effect is ascribed to the competition between the UAG-translating tRNA and RF-1 [34], it is quite reasonable that our present cell-free systems from the RF-1-free *E. coli* strains are not restricted to the context rule. Thus, the RFzero-iy-based cell-free system has the advantage that the positions for UAG codon insertion can be selected without consideration of the codon context effect, for highly efficient protein production.

Our RFzero-iy-based cell-free strategy is flexible and expandable. Once a cell extract is prepared, various non-natural amino acids could be incorporated, simply by supplementing the orthogonal aaRS and tRNA pairs, without any genetic engineering. There is no need to consider the effects on the growth of *E. coli*, which often matters during the in vivo incorporation of non-natural amino acids. We successfully incorporated acetyllysine at four positions in histone H4 [23], due to the efficient incorporation rate of the RFzero-iy-based cell-free system. Recently, we actually used the present RFzero-iy-based cell-free system for the structural analysis of a tetra-acetylated nucleosome core particle [35]. Therefore, we believe that our RFzero-iy-based cell-free strategy is useful to synthesize proteins with multiple, site-specifically incorporated non-natural amino acids (Figure 1).

In previous studies of the site-specific incorporations of non-natural amino acids, several RF-1 deleted *E. coli* strains, with numerous replacements of UAG codons to the other stop codons, were shown to successfully improve the incorporation efficiencies. Likewise, several cell-free protein synthesis methods using the cell extracts from these strains exhibited high incorporation efficiencies [17,18,19,20]. Meanwhile, *E. coli* strains have been generated specifically for various reasons, and the demands to modify these *E. coli* genomes for the incorporation of non-natural amino acids will arise. Therefore, our method for generating the RFzero-iy strain, simply by transforming a BAC plasmid and deleting the *prfA* gene from the genome, which is more rapid and convenient than the other reported methods, represents a promising way to generate new RF-1-deleted *E. coli* strains from pre-existing strains with genomes modified for specific purposes. Using a single type of cell extract, we could incorporate several non-natural amino acids, some of which were incorporated with low efficiencies upon in vivo expression. Cell-free protein synthesis has the potential to incorporate non-natural amino acids that are difficult to incorporate by in vivo expression, due to their toxicities and low cellular uptakes. Taken together, the advantages of protein synthesis with the present RFzero-iy-based cell-free strategy will expand the potentials of diverse *E. coli* strains for non-natural amino acid incorporation, and it will be a promising technique for advanced protein engineering.

## 4. Materials and Methods

### 4.1. Strains and Plasmids

*Escherichia coli* BW25113 was provided by the National BioResource Project (Japan), and BL21(DE3) was purchased from Novagen (USA). The BW25113-based and BL21(DE3)-based RFzero-iy strains were generated as described previously [8]. For the expression of chloramphenicol acetyltransferase (CAT), pk7-CAT [36] was used. For the expression of a superfolder-type GFP mutant, pN11-GFPS2, a plasmid encoding GFPS2 (GenBank: LC185343.1), which contains a poly-histidine affinity tag (N11, MKDHLIHNHHKHEHAHAEH), a protease site, and a GFPS1 [27] variant gene, in the pCR2.1 vector (Thermo Fisher Scientific, Waltham, MA, USA), was used. For the incorporation of non-natural amino acids, the tyrosine codons at positions 21, 88, 192, 231, and 249 in pN11-GFPS2 were individually replaced with a UAG codon, using a QuikChange Site-Directed Mutagenesis kit (Agilent Technologies, Santa Clara, CA, USA). In addition to the Y88amb mutation, the plasmid pN11-GFPS2(Y88amb) had the Y249C mutation, due to a PCR error.

### 4.2. Non-Natural Amino Acids

The non-natural amino acids were purchased from the following companies: 3-Iodo-l-tyrosine (IY) from Sigma–Aldrich (St. Louis, MO, USA) and Tokyo Chemical Industry (Tokyo, Japan), 4-Azido-l-phenylalanine (AzF), *p*-benzoyl-l-phenylalanine (*p*Bpa), and *N*^ε^-benzyloxycarbonyl-l-lysine (ZLys) from Bachem (Switzerland), *O*-Sulfo-l-tyrosine (Tys) from Watanabe Chemical Industries (Hiroshima, Japan), and *N*^ε^- (*m*-azidobenzyloxycarbonyl)-l-lysine (*m*AzZLys) from Sundia (Shanghai, China).

### 4.3. Cell Culture

To examine the growth profiles, the cells were grown at 37 °C in 5 mL of LB medium or 2YT medium (16 g tryptone, 10 g yeast extract, 5 g NaCl per liter). To the media, chloramphenicol (Cm) was added at 10 mg/L, and IY was added at 0.1 mg/mL. All of the culture tubes were shaken at 200 rpm.

To prepare the S30 cell extracts, the cells were preliminarily grown at 37 °C overnight in 200 mL of 2YT medium supplemented with 10 mg/L Cm and 0.1 mg/mL IY, in 1-liter or 2-liter baffled flasks shaken at 110–120 rpm.

For preparing S30 cell extracts with baffled flasks, the overnight culture was added to 2 L of 2YT medium to an initial cell density higher than 0.1 OD_600_, with 10 mg/L Cm and 0.1 mg/mL IY in a 5-liter baffled flask, and incubated at 30 °C with continuous shaking at 110–120 rpm. The cells were harvested at the late-log phase.

For preparing S30 cell extracts with a jar fermenter, the overnight culture was added to 16 L of 2YT medium with 10mg/L Cm and 0.1mg/mL IY in a 30-liter fermenter (37 °C, 220 rpm), to an initial cell density higher than 0.1 OD_600_. The cells were harvested at the late-log phase.

### 4.4. S30 Preparation

The S30 cell extracts were prepared as described previously [26,27]. The cells were harvested at the late-log phase and then washed three times with S30 buffer A (10 mM Tris-acetate buffer (pH 8.2) containing 14 mM Mg(OAc)_2_, 60 mM potassium acetate, and 2 mM DTT). The washed cells were suspended in S30 buffer B (S30 buffer A with 1 mM DTT) and crushed with glass beads (*ø* = 0.17–0.18 mm, B. Braun, Hessen, Germany), using a multi-bead shocker (Yasui Kikai, Osaka, Japan) three times at 2700 rpm for 30 s. For treating 1 g of cells, 1.25 mL of S30 buffer B and 3.15 g of glass beads were used. The cell debris and glass beads were removed by two rounds of centrifugation at 30,000× *g* for 30 min at 4 °C. The obtained supernatant was mixed with a 0.3-fold volume of pre-incubation buffer (293 mM Tris-acetate buffer (pH 8.2) containing 9 mM Mg(OAc)_2_, 13.2 mM ATP, 84 mM phosphoenolpyruvate, 4.4 mM DTT, 20 amino acids (40 M each), and 6.7 U/mL pyruvate kinase), and incubated for 70–80 min at 37 °C. The crude extract was dialyzed three or four times against a 50-fold volume of S30 buffer B, for 45–60 min at 4 °C. The cell extract was then centrifuged at 4000× *g* for 10 min at 4 °C, to obtain the supernatant as the S30 cell extract. Aliquots of the extract were immediately frozen in liquid nitrogen, and stored in liquid nitrogen or below −80 °C.

For preparing S30 cell extracts with a jar fermenter, the cells were harvested at the late-log phase and washed three times in S30 buffer B. The washed cells were suspended in the corresponding volume of S30 buffer B per weight (1 mL per 1 g cells) and crushed with a high-pressure homogenizer (Stansted, UK), which was operated at 170–200 MPa. The cell debris was removed by two rounds of centrifugation at 30,000× *g* for 30 min at 4 °C. The following procedures were conducted according to the baffled flask method.

### 4.5. Cell-Free Protein Synthesis

The small-scale dialysis-mode cell-free protein synthesis with a 30 µl reaction mixture was performed as described previously [37]. Chloramphenicol acetyltransferase (CAT) synthesis was performed at 30 °C. Synthesis of a superfolder-type GFP mutant, GFPS2 (GenBank: LC185343.1) was performed at 25 °C, without polyethlene glycol. All of the reactions were shaken at 240 rpm for 4 h. IY was supplemented at 1.0 mM in both the reaction and external solutions. Other tyrosine derivatives were supplemented at 1.0 mM to both solutions, with their specific TyrRS variants added to the reaction solution at 0.3 µg/µl. Lysine derivatives were supplemented at 1.0 mM to both solutions, with a tRNA^Pyl^ and a pyrrolysyl-tRNA synthetase variant, PylRS(R61K−Y384F−Y306A) [30], added to the reaction solution at 10 µM each.

### 4.6. Detection of Protein Synthesis

For the SDS-PAGE analysis, a 5-μL aliquot of the reaction solution was centrifuged, and the insoluble fractions were suspended in SDS sample buffer. The supernatants were mixed with 40 μL ice-cold 75% acetone, incubated at −20 °C for 30 min, and centrifuged to remove the polyethylene glycol. The precipitated proteins were then suspended in SDS sample buffer, as the soluble fraction. SDS-PAGE was performed on a 2–20% gradient gel, which was then stained with Coomassie Brilliant Blue. The GFPS2 fluorescence was measured using a fluorescence plate reader, ARVO SX (PerkinElmer, Waltham, MA, USA), with excitation at 485 nm and emission at 535 nm. The purified wild-type GFPS2 was used as the standard to calculate the productivity.

## Figures and Tables

**Figure 1 ijms-20-00492-f001:**
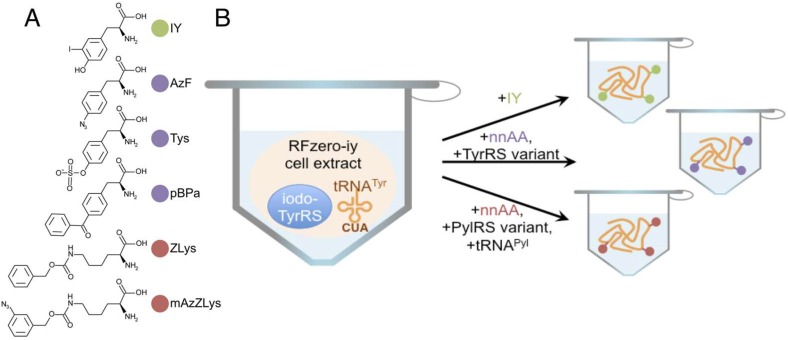
Incorporation of various non-natural amino acids by cell-free protein synthesis. (**A**) Structures of the non-natural amino acids used in this study. Colors indicate the types of non-natural amino acids shown in (**B**). (**B**) The concept of the RFzero-iy-based cell-free system for the incorporation of various non-natural amino acids, using a single type of cell extract. This strategy could be expanded to incorporate a variety of non-natural amino acids, by supplementing TyrRS variants, or tRNA^Pyl^ and PylRS variant pairs.

**Figure 2 ijms-20-00492-f002:**
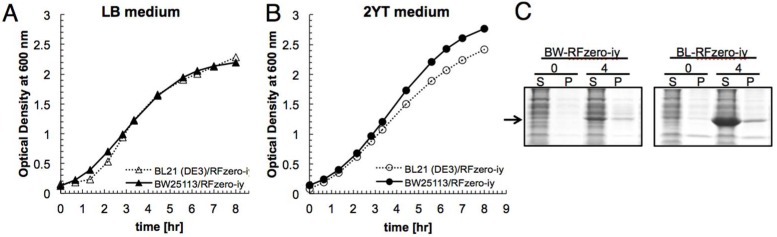
Comparison of RFzero strains derived from *E. coli* K- and B-strains as sources of S30 cell extracts. The growth profiles of the BW25113- and BL21(DE3)-based RFzero-iy strains in LB (**A**) and 2YT (**B**) at 37 °C, and the cell-free protein synthesis of Chloramphenicol acetyltransferase (CAT) with the S30 extracts from these strains (**C**). Aliquots of the reaction solution from the cell-free protein synthesis were taken at 0 and 4 h, and analyzed by SDS-PAGE with Coomassie Brilliant Blue staining. “S” and “P” indicate the soluble and insoluble fractions, respectively. The arrow indicates the position of CAT.

**Figure 3 ijms-20-00492-f003:**
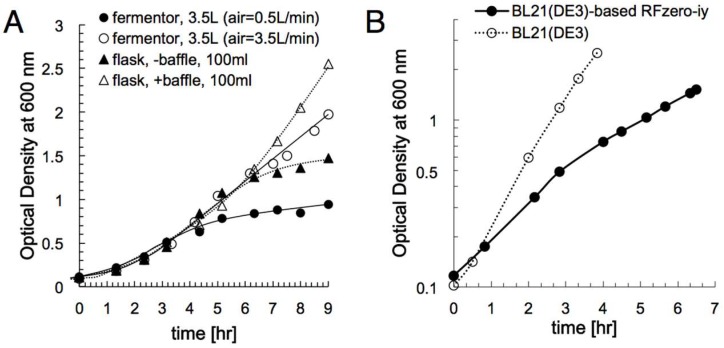
Growth profiles of RFzero-iy and BL21(DE3). (**A**) BL21(DE3)-based RFzero-iy was grown in a standard flask (filled triangles), a baffled flask (open triangles), and a jar fermenter with air supply rates of 0.5 L/min (filled circles) and 3.5 L/min (open circles). (**B**) BL21(DE3) (open circles) and BL21(DE3)-based RFzero-iy (filled circles), grown in jar fermenters at 30 °C.

**Figure 4 ijms-20-00492-f004:**
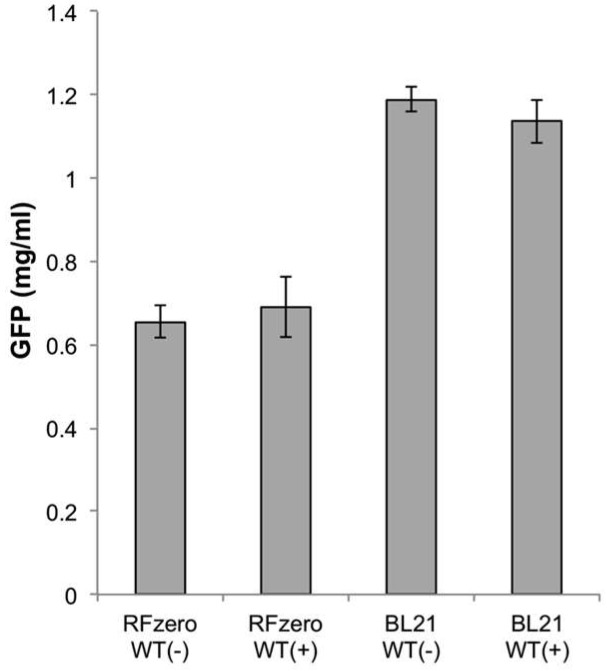
Protein productivity of RFzero-iy and BL21(DE3)-based S30 extracts. The cell-free protein synthesis of wild type GFPS2 was conducted with (+) or without (−) IY supplementation, using RFzero-iy-based S30 extracts (RFzero WT) and BL21(DE3)-based S30 extracts (BL21 WT).

**Figure 5 ijms-20-00492-f005:**
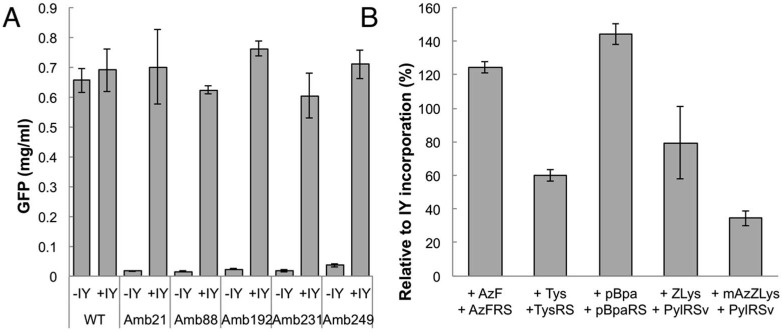
RFzero-iy-based cell-free system for non-natural amino acid incorporation. (**A**) The cell-free protein synthesis of wild-type N11-GFPS2 and UAG codon incorporated N11-GFPS2 mutants, with (+) or without (−) IY supplementation, were conducted with RFzero-iy-based S30 extracts. Wild-type GFPS2 is indicated as “WT”, and GFPS2 variants with UAG codons are indicated with the position numbers indicated after the prefix “Amb”. (**B**) The cell-free protein synthesis of N11-GFPS2(Y21amb) with supplementations of various non-natural amino acids and aaRS pairs, using RFzero-iy-based S30 extracts. PylRS(R61K–Y384F–Y306A) variant is indicated as PylRSv. Values are relative to the GFPS2 productivity of IY incorporation.

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
