# Peer review of "Cell-Free Protein Synthesis Using S30 Extracts from Escherichia coli RFzero Strains for Efficient Incorporation of Non-Natural Amino Acids into Proteins"

_ijms, 2019, doi:10.3390/ijms20030492_

Round 1

Reviewer 1 Report

Adachi et al. present interesting experimental results on an improved protocol for setting the cell-free protein synthesis preparations. The presented protocol using a RFzero-iy strain is demonstrated to effectively allow the desired protein synthesis by assigning to UAG translation to non-natural amino acids and removing the RF-1 proteins of the preparations.

Despite the novelty and interesting topic covered by the study, some minor issues need to be addressed before it can be accepted for publication:

First paragraph of the Results (lines 110-112): this is the description of the Results section, I mean, it is the instruction for writing the Results section. I think someone forgot to remove it before submitting the manuscript to the journal.

Results line 126: change "compered" to "compared", it is probably a typo.

Fig. 2 could be improved in quality (higher resolution) as the dots look blurry. Also, it would be good to include the symbols as legends to the panels. Figure 1, 3 and 4 include proper annotation of the symbols and the charts.

Author Response

Response to Reviewer 1 Comments

Comment: Adachi et al. present interesting experimental results on an improved protocol for setting the cell-free protein synthesis preparations. The presented protocol using a RFzero-iy strain is demonstrated to effectively allow the desired protein synthesis by assigning to UAG translation to non-natural amino acids and removing the RF-1 proteins of the preparations.

Response: We are grateful to the reviewer for the favorable comments.

Comment: Despite the novelty and interesting topic covered by the study, some minor issues need to be addressed before it can be accepted for publication:

First paragraph of the Results (lines 110-112): this is the description of the Results section, I mean, it is the instruction for writing the Results section. I think someone forgot to remove it before submitting the manuscript to the journal.

Response: We have removed the paragraph as requested.

Comment:  Results line 126: change "compered" to "compared", it is probably a typo.

Response: We have corrected the typo.

Comment: Fig. 2 could be improved in quality (higher resolution) as the dots look blurry. Also, it would be good to include the symbols as legends to the panels. Figure 1, 3 and 4 include proper annotation of the symbols and the charts.

Response: We have replaced the original Figure 2 with a higher resolution figure, and added the symbols to the panels as recommended.

Reviewer 2 Report

Authors, based on already published methods, generated two types of RFzero-iy strains from E.coli K-12, BW25113 and E. coli B strain, BL21(DE3). They compared the growth rates of both strains in LB and 2YT media. Next, they tested protein production in the extracts obtained from both strains. BL21(DE3)-based RFzero-iy S30 extract allowed for larger amount of protein production and was chosen for subsequent studies. Authors optimized aeration conditions for optimal growth of the cell culture. Different types of flasks (standard and baffled bottom) and jar fermenter airflow rates were tested. Prepared extract was tested for protein productivity. Wild-type GFPS2 and GFPS2 with tyrosine codons replaced with UAG were synthesized in cell-free lysates in the presence and absence of IY. Finally, authors tested different non-natural amino acids incorporation by supplementing cell-free lysates with specific aaRS variant and UAG-reading tRNA pair.

Presented experiments are correctly designed, performed and described. However, I would like that the authors better highlighted the novelty of the work. Especially since E. coli strain engineered to lack the release factor 1 (RF-1) was already published.

I also noticed several editorial errors in the manuscripts:

Lines 90-92 – Phrases are hard to understand,

Lines 110 – 112 – should be removed,

Figure 1 – the arrow is missing,

In line 120 authors mention three hours harvest point but in the description of the figure 1 it is four hours harvest point,

I think that figure 5 should be placed in the beginning rather than at the end of the manuscript.

Author Response

Response to Reviewer 2 Comments

Comment: Authors, based on already published methods, generated two types of RFzero- iy strains from E.coli K-12, BW25113 and E. coli B strain, BL21(DE3). They compared the growth rates of both strains in LB and 2YT media. Next, they tested protein production in the extracts obtained from both strains. BL21(DE3)-based RFzero-iy S30 extract allowed for larger amount of protein production and was chosen for subsequent studies. Authors optimized aeration conditions for optimal growth of the cell culture. Different types of flasks (standard and baffled bottom) and jar fermenter airflow rates were tested. Prepared extract was tested for protein productivity. Wild-type GFPS2 and GFPS2 with tyrosine codons replaced with UAG were synthesized in cell-free lysates in the presence and absence of IY. Finally, authors tested different non- natural amino acids incorporation by supplementing cell-free lysates with specific aaRS variant and UAG-reading tRNA pair.

Response: We are grateful to the reviewer for the positive comments.

Comment: Presented experiments are correctly designed, performed and described. However, I would like that the authors better highlighted the novelty of the work. Especially since E. coli strain engineered to lack the release factor 1 (RF-1) was already published.

Response: We have added explanations to highlight the novelty of the work in the introduction (lines 125-129) and discussion (lines 289-294) sections. We have switched Figure 5 to the new Figure 1, and we have mentioned the advantages of this system relative to the previously published RF-1-free E. coli strains, and the benefits of using cell-free protein synthesis for in vivo expression, using extract from RFzero strains.

Comment: I also noticed several editorial errors in the manuscripts:

Lines 90-92 – Phrases are hard to understand,

Response: We have divided and changed the phrases (lines 91-93) as recommended.

Comment: Lines 110 – 112 – should be removed,

Response: We have removed the lines as commented.

Comment: Figure 1 – the arrow is missing,

Response: We have added the arrow as commented.

Comment: In line 120 authors mention three hours harvest point but in the description of the figure 1 it is four hours harvest point,

Response: “Three hours” is the length of the E. coli cell culture for the cell extract preparation. “Four hours” is the length of the CAT synthesis by cell-free protein synthesis. To prevent confusion, we have changed “cell-free reaction solution” to “reaction solution from cell-free protein synthesis” in Figure 1.

Comment: I think that figure 5 should be placed in the beginning rather than at the end of the manuscript.

Response: We have moved Figure 5 to Figure 1, and have added the structures of the amino acids for an additional explanation, as recommended.

Reviewer 3 Report

The authors have presented a protocol for establishing a RFzero-iy-based cell-free system, which enables incorporation of a variety of non-natural amino acids into multiple sites of proteins at a high incorporation rate. The protocol would be helpful for researchers who want to charge non-natural amino acids into proteins. However, I have several questions and comments. So, the authors should answer my questions and comments as described below.

Major corrections

   The authors describe in Introduction that more than 100 non-natural amino acids have already been site-specifically incorporated into proteins for various purposes, such as conjugations with fluorescent probes, polymers, and drugs. In addition, the authors also describe that several RF-1-free E. coli strains have developed by deleting the RF-1-encoding gene prfA from genomic DNA. Therefore, it seems to me that the authors only present the precise strategy for site-specific incorporation of non-natural amino acids using the RFzero extract-based cell-free system, of which preparation is rapid and simple. However, I consider that the purpose of such a technique incorporating non-natural amino acids into proteins should be not always to provide experimental methods for incorporating non-natural amino acids into proteins, but, for example, to improve functions of natural proteins by incorporation of non-natural amino acids, and to explore significance of natural 20 amino acids and possibilities of the genetic code expansion and so on.

   So, the authors first should reply to my questions and comments described below.

1. What new answers to questions, which could not be given by previous similar strategies, could be provided by using the techniques presented in this paper? I consider that significance of this paper is too low to publish in IJMS, if the authors cannot give appropriate answers to the question.

2. Similarly, the results given in Fig. 4A indicate that incorporation of IY into five sites of GFP2 does not show any big effect to the protein. Therefore, it should be described about significance and purposes of the incorporation of non-natural amino acid, especially IY, into proteins, because it is supposed that effect of IY incorporation into other proteins should be also small.  

3. The vertical axis of Fig. 4b is “Relative to IY Incorporation (%)”. It is supposed from Detection of protein synthesis in Method that the relative incorporation was measured by fluorescent intensity of GFP synthesized with the respective cell-free protein syntheses. However, the fluorescent intensity should be affected by not only the activity of aaRSs with a non-natural amino acid, but also incorporation rate of the non-natural amino acid into GFP or the amount of GFP synthesized and the specific fluorescent intensity of GFP, into which non-natural amino acid was incorporated. Therefore, the authors should explain the reason why the values of “Relative to IY Incorporation (%)” change every non-natural amino acid used in Fig. 4b in more detail. 

4. It is difficult from the explanation of Fig. 5 to understand the reason why the RFzero-iy-based cell-free system is used for incorporation of non-natural amino acids into proteins, because the system enables only IY incorporation, if TyrRS variant and PylRS variant do not use. Therefore, the authors should explain significance of the RFzero-iy-based cell-free system for incorporation of non-natural amino acids in more detail.

Minor corrections

1. Figs. 1A and 1B should be exchanged, because the order of the figures is different from that described in the text.

2. The arrow indicating the position of CAT is not drawn in Fig. 1C.

Author Response

Response to Reviewer 3 Comments

Comment: The authors have presented a protocol for establishing a RFzero-iy-based cell- free system, which enables incorporation of a variety of non-natural amino acids into multiple sites of proteins at a high incorporation rate. The protocol would be helpful for researchers who want to charge non-natural amino acids into proteins. However, I have several questions and comments. So, the authors should answer my questions and comments as described below.

Major corrections

The authors describe in Introduction that more than 100 non-natural amino acids have already been site-specifically incorporated into proteins for various purposes, such as conjugations with fluorescent probes, polymers, and drugs. In addition, the authors also describe that several RF-1-free E. coli strains have developed by deleting the RF-1-encoding gene prfA from genomic DNA. Therefore, it seems to me that the authors only present the precise strategy for site-specific incorporation of non-natural amino acids using the RFzero extract- based cell-free system, of which preparation is rapid and simple. However, I consider that the purpose of such a technique incorporating non-natural amino acids into proteins should be not always to provide experimental methods for incorporating non-natural amino acids into proteins, but, for example, to improve functions of natural proteins by incorporation of non-natural amino acids, and to explore significance of natural 20 amino acids and possibilities of the genetic code expansion and so on.

So, the authors first should reply to my questions and comments described below.

 Response: We are grateful to the reviewer for the detailed comments.

Comment 1: What new answers to questions, which could not be given by previous similar strategies, could be provided by using the techniques presented in this paper? I consider that significance of this paper is too low to publish in IJMS, if the authors cannot give appropriate answers to the question.

Response 1: The significance of this strategy for the incorporation of non-natural amino acids is that it will expand the opportunities to use other E. coli strains, which have been generated for specific purposes in various research fields. To utilize the function of these generated E. coli strains for non-natural amino acid incorporation, the previously reported method to create the RF-free E. coli strains required extensive gene editing. In contrast, the method for generating the RFzero-iy strain was simple, by transforming a BAC plasmid and deleting the single prfA gene from the genome of any E. coli strain. Using the RFzero-iy strain, other non-natural amino acids could be incorporated into proteins in vivo, after supplementing the medium with specific non-natural amino acids during the culture, although these non-natural amino acids showed lower protein productivities as compared to iodo-tyrosine (IY).

This paper employs the cell-free protein synthesis with cell extracts from RFzero-iy strains. Using a single type of cell extract, various non-natural amino acids can be incorporated into proteins with excellent protein productivities. This system is not restricted to the use of TyrRS variants, but it is applicable to PylRS variants, which exhibit unusually broad specificities. This method is quite simple. Thus, it will expand the potentials of diverse E. coli strains for the incorporation of non-natural amino acids, and it will be a promising technique for advanced protein engineering.

We have added explanations to highlight the novelty of the work, the advantages of this system relative to the previously published RF-1-free E. coli strains, and the benefits of using cell-free protein synthesis for in vivo expression, using RFzero strains, in the introduction (lines 125-129) and discussion (lines 289-294) sections.

Comment 2: Similarly, the results given in Fig. 4A indicate that incorporation of IY into five sites of GFP2 does not show any big effect to the protein. Therefore, it should be described about significance and purposes of the incorporation of non- natural amino acid, especially IY, into proteins, because it is supposed that effect of IY incorporation into other proteins should be also small.

Response 2: The main purpose of Figure 5A (former Figure 4A) was to demonstrate the effectiveness of this system for the highly productive incorporation of non-natural amino acids into proteins, regardless of the position in the amino acid sequence. To emphasis this point, we have added the explanation in line 220. Taking account of your comment, we have added the explanations of the IY properties in lines 212-215, such as labeling proteins and conferring the heat resistance of proteins, and for X-ray crystallographic analyses.

Comment 3: The vertical axis of Fig. 4b is “Relative to IY Incorporation (%)”. It is supposed from Detection of protein synthesis in Method that the relative incorporation was measured by fluorescent intensity of GFP synthesized with the respective cell-free protein syntheses. However, the fluorescent intensity should be affected by not only the activity of aaRSs with a non-natural amino acid, but also incorporation rate of the non-natural amino acid into GFP or the amount of GFP synthesized and the specific fluorescent intensity of GFP, into which non-natural amino acid was incorporated. Therefore, the authors should explain the reason why the values of “Relative to IY Incorporation (%)” change every non-natural amino acid used in Fig. 4b in more detail.

Response 3: The effects of the incorporated non-natural amino acids on the fluorescent intensity are presumed to be small, because the incorporation site Y21 is located in the linker sequence of pN11-GFPS2, outside of the GFP. As you pointed out, the incorporation rates of non-natural amino acids may vary during translation, and the aminoacylation activities of aaRSs vary with each of the non-natural amino acids. We have added this explanation about Figure 5B (former Figure 4B) in lines 258-265.

Comment 4: It is difficult from the explanation of Fig. 5 to understand the reason why the RFzero-iy-based cell-free system is used for incorporation of non-natural amino acids into proteins, because the system enables only IY incorporation, if TyrRS variant and PylRS variant do not use. Therefore, the authors should explain significance of the RFzero-iy-based cell-free system for incorporation of non-natural amino acids in more detail.

Response 4: We have added explanations about Figure 1(former Figure 5) to explain the system to incorporate various non-natural amino acids using a single type of cell extract, by supplementing TyrRS variants or tRNAPyl/PylRS variants. Individual cell extracts are not required, and thus additional genetic engineering is not required. This is a simple and rapid property for the incorporation of various non-natural amino acids. We have also added an additional explanation after the corresponding explanation to the introduction section in lines 125-129, and to the discussion section in lines 290-294.

Comment: Minor corrections

(1)       Figs. 1A and 1B should be exchanged, because the order of the figures is different from that described in the text

(2)       The arrow indicating the position of CAT is not drawn in Fig. 1C.

Response 5: We corrected the errors as follows.

(1)       We have exchanged Figures 1A and 1B as commented.

(2)       We have added the arrow in Figure 1C as commented.

Round 2

Reviewer 3 Report

I have confirmed that authors have appropriately revised and corrected the previous manuscript following my comments and suggestions.